# The Role of Metabolism in Tumor Immune Evasion: Novel Approaches to Improve Immunotherapy

**DOI:** 10.3390/biomedicines9040361

**Published:** 2021-03-31

**Authors:** Alberto Cruz-Bermúdez, Raquel Laza-Briviesca, Marta Casarrubios, Belén Sierra-Rodero, Mariano Provencio

**Affiliations:** 1Medical Oncology Department, Health Research Institute Puerta de Hierro–Segovia de Arana (IDIPHISA) & Puerta de Hierro Hospital, Manuel de Falla Street #1, 28222 Madrid, Spain; rlaza@idiphim.org (R.L.-B.); mcasarrubios@idiphim.org (M.C.); bsierra@idiphim.org (B.S.-R.); 2PhD Programme in Molecular Biosciences, Faculty of Medicine Doctoral School, Universidad Autónoma de Madrid, 28222 Madrid, Spain

**Keywords:** immunotherapy, metabolism, immune evasion, new drugs

## Abstract

The tumor microenvironment exhibits altered metabolic properties as a consequence of the needs of tumor cells, the natural selection of the most adapted clones, and the selfish relationship with other cell types. Beyond its role in supporting uncontrolled tumor growth, through energy and building materials obtention, metabolism is a key element controlling tumor immune evasion. Immunotherapy has revolutionized the treatment of cancer, being the first line of treatment for multiple types of malignancies. However, many patients either do not benefit from immunotherapy or eventually relapse. In this review we overview the immunoediting process with a focus on the metabolism-related elements that are responsible for increased immune evasion, either through reduced immunogenicity or increased resistance of tumor cells to the apoptotic action of immune cells. Finally, we describe the main molecules to modulate these immune evasion processes through the control of the metabolic microenvironment as well as their clinical developmental status.

## 1. Introduction

### 1.1. Tumor Microenvironment and Metabolism

One of the key hallmarks of cancer is its altered energy metabolism. Uncontrolled tumor growth requires both energy and building materials, for which neoplastic cells must consume and metabolize extracellular nutrients for ATP generation, de novo synthesis of macromolecules, and the maintenance of redox balance [1].

In contrast to differentiated cells, which in the presence of oxygen rely on the oxidative phosphorylation system (OXPHOS), tumor cells have the ability to reprogram their metabolism to maintain a highly glycolytic metabolism regardless of the presence of oxygen. This phenomenon of aerobic glycolysis characteristic of tumor tissues, described at the beginning of the 20th century, is known as the Warburg effect [2,3].

However, tumors are not uniform masses of cancer cells. There are different cell types, such as fibroblasts, endothelial cells, and immune cells (T and B cells; Natural killer cells, NKs; natural killer T cells, NKTs; myeloid-derived suppressor cells, MDSCs; regulatory T cells, Tregs, and tumor-associated macrophages, TAMs) among others, that cohabit with each other in the tumor microenvironment. Even within a cell type, not all cells are the same due to the accumulation of mutations and epigenetic changes. Finally, these cells modify their metabolism in response to different conditions in the microenvironment, such as nutrient availability, hypoxia, pH levels, waste products, or drugs present. These different cells compete and interact in the microenvironment, adding great complexity to the classical view of tumor metabolism [4,5,6,7,8] 

### 1.2. Immunoediting

The theory of cancer immunoediting was proposed in 2002 as an evolution of the concept of cancer immunosurveillance. This new, broader and integrating concept assumes that the immune system not only protects the body against tumor development, but can also sculpt the phenotype of a developing tumor to a point where tumor cells escape this control [9]. The ability of tumor cells to evade the immune system is currently one of the “hallmarks of cancer” [1].

Cancer immunoediting is divided into three phases, known as the three “E’s”: Elimination (immunosurveillance), equilibrium (resting state), and evasion (tumor escape). Although this process can be divided into phases for a better understanding, immunoediting must be understood as a continuum that can flow between the different phases. In this way it is possible to explain the influence of external factors (such as treatments, tumor microenvironment metabolism, or age-related deterioration of the immune system), which can modify both the speed of transition from one phase to another and the direction of this flow.

The process of cancer immunoediting occurs during the natural progression of tumors, but the available evidence from studies of patients treated with oncological immunotherapies indicates that this process is repeated, in part or in whole, in response to treatment [10]. Thus, tumors that are, by definition, in a tumor evasion phase, and after treatment with immunotherapy may return to an equilibrium phase (which would correspond to stable disease after treatment) or even return to an elimination phase (which would correspond to varying degrees of response).

#### 1.2.1. Elimination

This first phase is equivalent to the concept of cancer immunosurveillance. During the elimination phase, the innate and adaptive immune systems cooperate to recognize and kill transformed cells that have escaped the intrinsic mechanisms of tumor suppression.

During the development of the elimination phase different elements are involved including: cytokines (IFNα/β, IFN-γ, IL-1α/β, IL-2, IL-6, IL-12, and TNF), innate immunity cells (NK, NKT, Tγδ cells, dendritic cells, and macrophages); adaptive immunity cells (CD4+ and CD8+ T cells), and immune effector molecules (FasL, TRAIL, and Perforin) [9,10,11].

Tumor cells can cause alterations in the environment. This situation leads to the induction of inflammatory signals and the recruitment in the first instance of cells of the innate immune system (NK, NKT, Tγδ cells, M1 macrophages and dendritic cells) [11]. On the other hand, the transformed cells further express stress-induced molecules (e.g., calreticulin or NKG2D MICA/B ligands), that favor their recognition by infiltrating lymphocytes (NK, NKT, Tγδ) that are stimulated to produce IFN-γ. Innate immune cells, such as macrophages (M1) and granulocytes, also contribute to antitumor immunity by secreting TNF-α, IL-1, IL-12, and reactive oxygen species. IFN-γ produced in the microenvironment exhibits pro-apoptotic, anti-proliferative, and anti-angiogenic activities thereby reducing tumor growth and blocking the generation of new vessels [12]. In addition, CXCL10, CXCL9, and CXCL11 production is induced in the tumor microenvironment, which recruits cytotoxic lymphocytes (CTLs) and reduces tumor angiogenesis, inducing further tumor death [13].

The cell death generated by the different mechanisms described above favors the processing of cellular debris by local dendritic cells, which, once mature, are directed to the local lymph nodes. Mature dendritic cells reduce their capacity to capture antigens and increase their antigen presentation capacity (increasing the expression of MHC molecules and adhesion and co-stimulation molecules, such as CD40, CD54, CD58, CD80, CD83, and CD86). Once in the lymph nodes, the dendritic cells activate tumor-specific CD4+ (Th1) T lymphocytes that produce IFN-γ. These cells, in turn, facilitate the development of tumor-specific cytotoxic CD8+ T cells [14].

The tumor-specific CD4+ and CD8+ T cells are directed to the tumor site, where cytotoxic T lymphocytes destroy tumor cells, whose immunogenicities have been enhanced by exposure to locally produced IFN-γ. CD8+ T cells can induce tumor cell apoptosis by secretion of perforin and granzymes, or by interacting with Fas and TRAIL receptors on tumor cells. In addition, effector T cells express co-stimulatory molecules, such as CD28, CD137, GITR, or OX-40, that enhance their proliferation and survival [15].

In the ideal situation of the elimination phase, a circle of tumor death, antigenic presentation and activation-recruitment of the immune system occurs, leading to the complete elimination of all transformed cells, ruling out the possibility of clinical manifestation of the disease.

#### 1.2.2. Equilibrium

This second phase is defined as the period during which the immune system and the cancer coexist in apparent harmony in the body. During this phase, although the immune system is not able to completely eliminate the tumor cells, it is able to keep their number stable, preventing the tumor from progressing in situ or generating new distant metastases.

This incomplete control exercised by the immune system generates a classic Darwinian selection on the tumor cells present, in which many of the variants are eliminated but others, with a greater resistance to immune attack, are selected. Thus, the equilibrium phase involves the continuous removal of tumor cells and the selection of the most resistant variants. The molecular mechanisms that trigger or maintain the equilibrium phase are poorly understood due to the difficulty of adequately modeling this stage. Importantly, the balance between IL-12 (which favors clearance) and IL-23, and to a lesser extent IL-10 (which favor tumor persistence), keeps tumors in equilibrium. However, other pathways, such as IL-4, IL-17A, TNF, and IFN-αβ, are dispensable to reach equilibrium. It remains to be determined the most relevant elements that shift this equilibrium towards tumor elimination or escape. In any case, oncologic patients present, by definition, tumors that have been edited during the equilibrium phase and are in the evasion phase [9,10,11].

#### 1.2.3. Evasion

The third stage of the model involves a situation in which the cancer is able to progress beyond the controls of the immune system. It is during this stage that the tumor cells become clinically relevant and the cancer can be diagnosed. Due to this, it is the phase for which we have the most information and, therefore, also where most therapeutic efforts have been focused [9,10,11].

There are numerous evasion mechanisms, potentially as many as there are factors controlling the immune system. When it comes to classifying them, they can be divided into elements that favor a loss of immunogenicity, understood as the loss of the capacity to generate an immune response, and those that favor greater resistance of the tumor cells.

The knowledge of these evasion mechanisms has allowed the development of therapies based on their control [16]. This is the case of anti-PD-1/PD-L1 or anti-CTLA-4 therapies that have revolutionized cancer treatments. Anti-PD1/PDL1 therapy is based on the blockade of PD-1 protein in lymphocytes or PD-L1 in tumor cells, preventing lymphocytes inactivation and promoting tumor elimination. Similarly, anti-CTLA-4 treatment is based on the blockade of CTLA-4 in T-cells. CTLA-4 competes with the T-cell co-stimulatory receptor CD28, for binding to B7-1/B7-2 in antigen presenting cells, therefore promoting T-cell anergy [17].

In any case, removing these “brakes” only works if this is the main mechanism leading to tumor evasion and there are no other additional secondary brakes.

As we will see in detail later in this article, metabolism is responsible for numerous immune evasion mechanisms that may, on the one hand, favor tumor progression and, on the other hand, may compromise the success of current immunotherapies. However, as for current anti-PDL1/anti-CTLA4 therapies, the knowledge of these mechanisms represents an opportunity for possible future immunotherapies based on the regulation of these metabolic processes.

## 2. Role of Metabolism in Tumor Immune Evasion

### 2.1. Reduction of Tumor Recognition by Immune Cells

#### 2.1.1. Antigen Presentation

Losing the capacity to present neoantigens through human leucocyte antigen (HLA) is one of the main mechanisms of tumor evasion. Loss of heterozygosity or even loss of both alleles of mayor histocompatibility complex (MHC) has been described in several types of cancer [18,19,20]. Interestingly, MHC expression is influenced by tumor metabolism. Cell culture conditions that forced respiration instead of anaerobic glycolysis leads to an increase expression of ERK5, which enhance MHC-I transcription [21]. However, cell lines lacking mitochondrial DNA, with reduced expression of MHC-I, showed similar sensitivity to CTL mediated killing but enhanced NK mediated killing [22]. Furthermore, it has been shown that hypoxic conditions downregulate MHC expression in a HIF-1α dependent manner, together with negative regulation of others antigen presenting proteins like TAP 1/2 and LMP7 [23]. Sometimes downregulation of MHC-I is not caused by genetic mutations but through preferential lysosomal degradation by an autophagy-dependent mechanism [24]. Recently, it has been described that combination of oxygen and glucose deprivation results in a decrease of antigen presentation by MHC I on cancer cells, through a mechanism that involves STAT1 phosphorylation and PI3K activity [25]. PI3K inhibitors also switched cancer cells to a more sensitive phenotype for recognition by CD8+ T cells.

In clinical samples of melanoma patients has been described a higher rate of oxidative and lipid metabolism in responders to anti-PD-1 immunotherapy. The authors identified that lipid metabolism was responsible for an increase in antigen presentation, followed by higher sensitivity to T cell-mediated killing [26].

#### 2.1.2. Regulation of Immune Checkpoint PD-L1

It has been described that hypoxic microenvironments upregulate PD-L1 expression on MDSCs, macrophages, dendritic cells and tumor cells by a direct binding of HIF family proteins to an hypoxia-response element (HRE) in the *PD-L1* proximal promoter [27,28]. Furthermore, hypoxia-induced expression of PD-L1 increased the resistance of tumor cells to CTL-mediated lysis [28], and its blockade enhanced T cell activation mediated by MDSCs, together with a reduced expression of MDSCs cytokines IL-6 and IL-10 [27].

Another metabolic pathway that regulates PD-L1 expression is glucose consumption. Enhanced glycolysis in tumors in enough to override the protective role of T cells to control tumor growth, and blocking PD-L1 decreases glycolysis by inhibiting mTOR activity and reduces expression of glycolysis enzymes [29]. More recently, it has been described that PD-L1 enhances glycolysis by upregulating hexokinase-2 (HK2) expression, the enzyme responsible of the glucose to glucose-6-phosphate (G6P) conversion. Consequently, tumors characterized by PD-L1^+^/HK2^high^ expression correlated with fewer presence of CD8+ T cells when compared to PD-L1^+^/HK2^low^ tumors [30].

Although cancer cells mainly employ glycolytic metabolism, resistant cells to cisplatin-based chemotherapy becomes more reliant on oxidative metabolism instead of glycolysis. This may lead to elevated levels of reactive oxygen species (ROS) in resistant tumors [31]. Elevated ROS and metabolic alteration drives to epithelial-mesenchymal transition (EMT), which ultimately leads to an increased expression on PD-L1 in these tumors [32].

Finally, NAD(+) metabolism participation in aging and cancer processes has been extensively investigated, but only very recently has it been shown that phosphoribosyltransferase (NAMPT), the rate-limiting enzyme of the NAD(+) biogenesis, induces PD-L1 by a IFNγ-dependent mechanism in multiple types of tumors [33]. Thus, high NAMPT expressing tumors are associated to a higher CD8+ T cell tumor immune evasion. However, this also imply enhanced efficacy of anti-PD-L1 antibody immunotherapy in these tumors, and opens the possibility of therapies based on NAD+ replenishment to sensitize anti-PD-L1 resistant tumors.

#### 2.1.3. Immunosuppressive Microenvironment

Hypoxia: Hypoxia in tumor microenvironment occurs when the pressure of oxygen drops lower than 510 mm Hg. This leads to an inadequate oxygen supply to cells and generates a chaotic tumor microvasculature network, that ultimately, fails to rectify the oxygen deficit. The hypoxia-inducible factor (HIF) family, specially HIF-1α, are transcription factors that under hypoxic conditions bind to the HRE in target genes and activates the expression of several molecules involved in various cellular pathways responsible for tumor resistance to several therapies, including immunotherapy [34,35,36].

The mechanisms by which hypoxia is capable of producing a suppression of the immune system are widely known: either by favoring the expression of PD-L1, V-Domain Ig Suppressor T Cell Activation (VISTA), and CD47 in hypoxic tumor cells, that inhibits T cells and prevents recognition by macrophages; as well as inducing autophagy [37,38] or MIC shedding [39]. Due to their special relevance, the role of lactate and adenosine levels will be explained in detail in the following sections.

VISTA expression is induced in hypoxic conditions and promotes the immunosuppressive functions of tumoral MDSCs [40], leading to the suppression of T cell proliferation and activity [41]. In addition, hypoxia upregulates the Macrophage Immune Checkpoint CD47 (commonly known as the “Don’t Eat Me” signal) inducing tumor cell escape from phagocytosis [42,43,44,45,46].

Autophagy is another cellular process implicated, in the way that the degradation of cellular components provides enough nutrients to cancer cells to maintain its cellular functions under stress conditions triggered by the hypoxic microenvironment [47]. Through several mechanisms, autophagy is capable of impairing tumor cell susceptibility to CTL and NK mediated killing [48,49,50,51,52].

Finally, several HREs have been identified in the promoter of the non-classical MHC-I *HLA-G* gene [53,54] that could be linked to its immunosuppressive functions and poor prognosis when expressed in various tumor types [55,56].

Lactate: Glycolysis is a less effective process compared to oxidative phosphorylation for ATP production. Thus, cancer cells tend to increase their glucose uptake and accumulate lactate as an extracellular component, leading to an acidification of the extracellular pH in tumor microenvironment, ranging between 6.0 and 6.5. The tumor microenvironment acidosis has been associated with a worse clinical prognosis, since it favors processes such as metastasis, angiogenesis and, more importantly, immunosuppression [57,58].

The acidic microenvironment generated by lactate production and release by cancer cells favors the recruitment of different immune cells that constitute a immunosuppressive microenvironment [59]. Related to innate immunity, it has been demonstrated that the presence of lactate induces the apoptosis on NK cells [60] and also dysregulates its cytolytic activity [61]. Moreover, high lactate environment is detrimental for NKT cell activation, survival and proliferation [62,63]. Additionally, it has been shown that lactate prevents the differentiation of dendritic cells and makes them tolerogenic, leading to the production of higher levels of IL-10 [64]. Regarding to macrophages, it has been described that a immunosuppressive M2 phenotype is induced by increased amounts of lactate [65,66].

Regarding adaptive immunity, it has been found that high levels of lactate inhibits T-cell proliferation and alters the cytokine production of cytotoxic lymphocytes [67], and further reduces the number and activity of CD8 T cells by inducing apoptosis [68]. Moreover, high expression of Lactate dehydrogenase A (LDHA), the enzyme responsible of the inter-conversion of lactate and pyruvate, is associated with lower expression of T cell markers [68] and its blockade effectively enhances the infiltration of CD8 T cells and improves the efficacy of antiPD-1 therapy in melanoma [69]. While this hypoxic environment with high levels of lactate reduces the function and activation of effector T cells, these same conditions support an adequate function of Tregs, which are able to consume lactate in symbiosis with tumor cells [6,70,71,72].

Adenosine: Adenosine in the tumor microenvironment deactivates the cytotoxic effector functions of NK and CD8+ T cells through A2A receptor (A2AR) signaling. In addition, adenosine polarizes myeloid cells to develop immunosuppressive phenotypes and increases the proliferation of Tregs and MDSCs [73,74]. The mechanism behind the immunosuppressive role of adenosine signaling is well studied. Briefly, A2AR trigger an increase of intracellular cAMP levels through stimulation of adenylyl cyclase. Increased intracellular cAMP levels activates protein kinase A, leading to several immunosuppressive effects, increasing the levels of TGF-beta, IL-10, PD-1, LAG-3, and Tregs, and decreasing the levels of pro-inflammatory cytokines, such as IFN-γ, TNF-α, and IL-6, among others [75,76,77]. Adenosine in the microenvironment is generated from extracellular ATP, through the consecutive activity of the ecto-nucleotidases CD39 and CD73 proteins, which are expressed on the membrane of tumor cells, as well as on stromal and immune suppressor cells, such as dendritic cells, Th17 lymphocytes, and M2 macrophages [78]. Adenosine may be further degraded to inosine by adenosine deaminase. Under normal conditions, the levels of extracellular ATP are insignificant; however, under certain situations of tissue stress (inflammatory processes, hypoxia, ischemia) common in the tumor microenvironment, there is a significant increase in these levels. This ATP release can be nonspecific or mediated by different mechanisms, including the participation of exocytotic granules, plasma membrane-derived microvesicles, specific ATP-binding cassette transporters, and membrane channels (connexin hemichannels, pannexin 1, CALHM1, volume-regulated anion channels, and maxi-anion channels) [77]. In parallel to increased ATP release, tumor microenvironment conditions may increase CD39 and CD73 expression through various inflammatory mediators, such as TGF-β, IFNs, TNF, IL-1β, and prostaglandin E2. In most tumor types (lung, colon, stomach, pancreas, among others) high levels of CD39 or CD73 expression are associated with a worse prognosis [75,78].

Glucose: Glucose availability acquires particular importance in T cell metabolism. In the steady state, T cells have low energetic demands regulated by the presence of interleukin-7 (IL-7), which promotes Glut1 trafficking of glucose [79]. On the one hand, memory T cells have greater mitochondrial mass and improved spare respiratory capacity, in part due to the need of developing a rapid recall response to antigen exposure [80]. On the other hand, the activation of T cells and the subsequent proliferation and differentiation to effector T cells requires large amounts of energy, achieved by the suppression of oxidative phosphorylation and activation of glycolysis, incorporating a higher amount of glucose [81,82]. Resting Tregs mainly use oxidative metabolism, but activation through Toll-like receptor (TLR) signals promote proliferation, glycolysis, and Glut1 expression via PI(3)K-AKT-mTORC1 signaling [83]. Subsequently, FOXP3 reprograms Treg metabolism by suppressing glycolysis and favoring a more oxidative phosphorylation phenotype [83,84,85].

Glutamine: In addition to glucose, high glutamine consumption is one of the hallmarks of cancer cells metabolism. Glutamine, the most abundant circulating amino acid in the human body, is converted to glutamate by the enzyme glutaminase (GLS) and internalized to the mitochondria to fuel TCA cycle for energy and intermediate metabolite production. Several studies have confirmed that high expression of GLS is correlated to high proliferative rates and malignancy in various cancer types [86], although it has also been described that GSL2 isoforms seem to have a tumor-suppressive function [87,88].

Moreover, glutamine is essential for ATP production and biosynthesis of nucleotides, proteins and lipids. Consequently, this amino acid possess a noteworthy role in T-cell proliferation and differentiation [89,90]. After activation, T cells exhibit an increase in glutamine uptake and metabolism [91,92]. Therefore, in microenvironments under glutamine deprivation, differentiation of T cells leads to a more immunosuppressive Treg phenotype [93]. Recently, it has been shown that effector T cells respond to glutamine antagonism by upregulating oxidative metabolism and conditioning infiltrating CD8+ T cells toward a highly proliferative, activated, and long-lived phenotype [94].

Tryptophan and arginine: A fundamental role of the catabolism of the essential amino acids tryptophan and arginine, in the tumor microenvironment in relation to immune escape, has been described. Most tumors overexpress Indoleamine 2,3-dioxygenase (IDO), this enzyme catalyzes the degradation of the essential amino acid tryptophan (Trp) to kynurenine (Kyn). T and NK cells are sensitive to IDO activity in the microenvironment, so when Trp levels are reduced by IDO activity they cannot multiply, leading to a reduced immune response. Additionally, this depletion in Trp and Kyn accumulation stimulates regulatory T cells and MDSCs promoting immunosuppression [95,96]. Thus, high IDO expression in the tumor or lymph nodes has been an adverse prognostic factor in melanoma, colon cancer, brain tumors and ovarian cancer, among others [97,98]. On the other hand, arginase 1 (Arg1) catalyzes the hydrolysis of the amino acid L-arginine to produce urea and L-ornithine, thus depleting extracellular L-arginine. This enzyme is expressed by all immunosuppressive myeloid cells (TAMs, MDSCs and granulocytes). T cells are auxotrophic for L-arginine, requiring this amino acid for rapid and successive rounds of proliferation during TCR-dependent activation. Depletion in L-arginine caused by its hydrolyzation results in the suppression of T-cell responses due to the decrease of the CD3ζ chain expression [99]. The presence of Arg1 is, therefore, associated with an inhibited or cold immune microenvironment [100].

Reactive oxygen species (ROS) and nitric oxide (NO): Reactive oxygen species (ROS) are mainly by-products of oxygen metabolism (OXPHOS system) or specific enzymatic reactions (nicotinamide adenine dinucleotide phosphate oxidase, NOX; dual oxidase, DUOX), that causes both damaging and beneficial effects depending on their balance in the microenvironment. ROS homeostasis is highly regulated by the antioxidative machinery including superoxide dismutase (SOD), catalase (CAT), and glutathione peroxidase (GPX). ROS and NO are responsible for multiple immune regulation processes related to tumor immune evasion [101,102].

In general terms, oxidative stress in the tumor microenvironment promotes immune evasion. It has been described how elevated levels of ROS are associated with decreased infiltration and activity of T and NK cells in tumors. Interestingly, a weak NRF2-associated antioxidant system and high vulnerability to free oxygen species has been described in Tregs. Apoptotic Tregs cells release and convert a large amount of ATP to adenosine via CD39 and CD73, and mediate immunosuppression via the adenosine and A2A pathway in the tumor microenvironment [103]. MDSCs and tumor-associated neutrophils inhibit lymphocytes functions through ROS production.

Moreover, control of microenvironment ROS levels strengthen the activation and activity of CAR-T cells that co-expressed catalase and bystander non-transfected effector cells [103]. Additionally, efficacy of adoptive T-cell therapy is improved by treatment with the antioxidant N-acetyl cysteine, which limits activation-induced T-cell death [104], reinforcing the key role of ROS homeostasis in tumor elimination.

NOS enzymes, using L-arginine as substrate, generate nitric oxide (NO). The inducible form of NOS (iNOS) is generally associated with the immune system and produces NO for prolonged periods in a calcium-independent manner. Beyond its role in carcinogenesis, including DNA damage and modulation of apoptosis, NO has a role in tumor evasion by inducing the activation and recruitment of MDSCs, Tregs and TAMs [105].

Fatty Acids: Fatty acids are relevant in the tumor microenvironment. On the one hand, they serve as a source of energy production and are essential elements of cell membranes, and on the other hand, they participate in signaling processes as second messengers [106]. Increased lipid levels in the tumor microenvironment may contribute to immune suppression by TAMs, MDSCs, and Treg cells. Tumor cells can stimulate lipid biosynthesis in human TAMs supporting their pro tumoral features, such as cytokine and ROS production [107]. Additionally, MDSCs go through a metabolic reprogramming from glycolysis to fatty acid oxidation, increasing fatty acid uptake, that is associated to their immunosuppressive activity on T-cells [108]. Although Tregs rely less on glucose consumption in comparison with T effector cells [109], recent data supports that this is not caused by an increased FAO but a dependence on lactic acid import to mitochondria [6].

### 2.2. Increase of Tumor Resistance

In addition to mechanisms favoring immune evasion through reduced immunogenicity or recognition, metabolism may influence other processes related to an intrinsic resistance of the tumor cells against lymphocyte activity.

#### 2.2.1. Decrease in Perforin/Granzyme Activity

Cytotoxic lymphocytes and NK cells have the property of production and secretion of toxins including the serine protease granzyme and the pore-forming protein Perforin, being this the major effector pathway over Fas/Fas-ligand interaction to induce cell death on tumor cells [110,111].

It has been described that increased autophagy in tumor cells, related to hypoxic microenvironment, is an immune evasion mechanism based on resistance to NK-mediated lysis. Hypoxic tumor cells present lower levels of NK cells derived granzyme, due to its degradation by autophagy during its intracellular trafficking, leading to cancer cell escape from NK cell action. Granzyme B (GZMB) enters target cells by endocytosis and traffic to enlarged endosomes allowing for the gradual release of this proteinand the initiation of apoptotic cell death [52]. Therefore, autophagy inhibition in this context induces tumor regression, facilitating the elimination of tumor cells by NK cell action through the release of perforin and granzyme [112].

#### 2.2.2. Decrease in IFN-γ Signaling

In addition to being a key factor in the immune antitumor response by regulating the activity of most immune cell types, IFN-γ exhibits antiproliferative and pro-apoptotic activities in tumor cells [12].

Through STAT1 activation, IFN-γ is able to, on the one hand, induce the expression of IFITM1, the cyclin-dependent kinases inhibitors p21 and p27 and the G1/S phase blocker KLF4 and, on the other hand, inhibit the expression of several cyclins and c-Myc and HER-2 [12,113]. Notably, hypoxia has a negative impact on *STAT1* transcription mediated by STRA13 [39]. Therefore, it is possible that a hypoxic microenvironment favors lower STAT1 levels in tumor cells that hinder antiproliferative IFN-γ signaling in the tumor cell. Additionally, IFN-γ also induces necroptosis in tumor cells through the activity of the serine–threonine kinase RIPK1 [12]. Similarly, the expression of RIPK1 is suppressed by hypoxia [114], which may participate as a mechanism of resistance of tumor cells to the action of IFN-γ in the tumor microenvironment. Recently, an immunostimulatory role of mitochondrial RNA in cytoplasm has been described, linking mitochondrial integrity to STAT1 phosphorylation and opening new therapeutic strategies to improve immunotherapy based on targeting mtDNA [115].

#### 2.2.3. Death Receptors

Cytotoxic lymphocytes and NK cells also induce death ligands, (tumor necrosis factor, TNF; Fas ligand, FasL, and TNF-related apoptosis-inducing ligand, TRAIL) to eliminate tumor cells through the activation of their specific receptors present on the tumor cells [116].

To our knowledge, no direct regulation of cell death ligands receptors (TNFR1, FAS, and DR4/5) by metabolism has been described. However, IFN-γ is able to further promote tumor cells apoptosis by increasing the levels of these receptors, as well as, by upregulating the expression of caspase-1, -3 and -8 [12].

#### 2.2.4. Decrease in Apoptotic Pathways

Finally, many of these processes depend, as last step, on the entry of the tumor cell into apoptosis. Metabolism is a key element in facilitating or hindering the process of apoptosis [117], being some of the pathways involved in this resistance mechanism to immunotherapy, common to other cytotoxic therapies, such as chemotherapies [53].

Cytotoxic lymphocytes and NK cells can trigger the extrinsic and intrinsic apoptosis pathways in tumor cells, using both caspase-dependent and -independent mechanisms. After death ligand receptor activation, DISC (death-inducing signaling complex) is ensembled and caspase 8 and 10 initiates a downstream signal cascade through caspases and BID (BH3 interacting domain death agonist) cleavage. Granzyme B initiates mitochondria-dependent apoptosis via BID, Mcl-1, BIM (BCL2 like 11), and p53 [118].

Mcl-1 is stabilized by high glucose metabolism through decreased GSK-3α/3β activity that prevents Mcl-1 phosphorylation and degradation [119]. Similarly, glucose metabolism also suppresses p53-mediated PUMA (p53-upregulated modulator of apoptosis) induction and apoptosis [120]. In addition, hypoxia suppresses expression of BID through HIF-1 and upregulates antiapoptotic BCL-2, BCLxL, and cIAP-2, as well as, downregulate proapoptotic BAX (BCL2 associated X, apoptosis regulator) and BAD (BCL2 associated agonist of cell death), independently of HIF-1 levels [121]. Finally, a role of mitochondrial mass and OXPHOS function in the initiation of apoptosis has been described, which could explain in part the relevance of mtDNA variants in cancer [31,122].

## 3. Therapeutic Approaches

These mechanisms by which metabolism favors the evasion of the immune system by tumors also open up the possibility of developing therapies based on the control of these processes. Table 1 and Table 2 list the main drugs in clinical or preclinical development, respectively.

IDO: No IDO or IDO1 inhibitors are currently approved for treatment though there are numerous studies testing their possible application in the clinical setting. The IDO inhibitors that are more advanced in clinical development are indoximod [123], epacadostat, and linrodostat mesylate [95]. Indoximod is being evaluated in up to 15 phase I, II and III trials in different tumors such as melanoma, glioma, NSCLC, pancreatic cancer and prostate cancer, both alone and in combination with different immuno-checkpoint agents. Epacadostat is an orally available hydroxyamidine that is being assessed in more than 70 clinical trials alone and in combination with pembrolizumab and nivolumab [124]. Linrodostat mesylate is also in advanced study in clinical trials, with up to 17 phase I, II, and III trials showing positive results in combination with nivolumab [125].

Other molecules assessed in clinical trials that target IDO are Navoximod, DN1406131, EOS200271 (PF-06840003) [126], KHK2455, LY01013, MK-7162, SHR9146 (HTI-1090). Additionally, several preclinical studies with novel molecules targeting not only IDO1 but also IDO2 and TDO are being evaluated in cancer as a single agent: DN016 [127], EPL-1410 [128], IACS-9779 [129], RG70099 [130], TQBWX220 [131], CMG017 [132], STB-C017 [133], and in combination with immunotherapy: DN016 [127], CMG017, STB-C017 [95,96].

Arginine: INCB001158 is an orally available inhibitor of arginase under clinical evaluation in solid tumors and multiple myeloma in several phase II trials: alone and in combination with chemotherapy and immunotherapy [134]. AB474 is an ARG1 inhibitor recently evaluated in pre-clinical studies [135]. Tadalafil, a PDE-5 inhibitor, is also being investigated in combination with PD-1 inhibitors as it seems to mediate immunostimulatory effects by reducing ARG1 and iNOS expression in MDSCs [100,136].

Lactate: The inhibition of LDHA, which expression is increased in tumors [137], is being investigated in pre-clinical setting with some agents: Oxamate [138], FX11 [139], NHI-I, NHI-2 [140], and 1,3-benzodioxole derivatives [141], which are competitive inhibitors of the enzyme lactate dehydrogenase. Regarding monocarboxylate transporters, inhibitors of MCT1 seem to be a promising new class of immunomodulatory drugs. These molecules reduce glycolytic rate of T lymphocytes and block their proliferation by inhibiting their acid lactic efflux. AZD3965 is currently under clinical development, and it is shown to inhibit the proliferation of multiple lymphoma cell lines [142]. BAY-8002 is also an MCT inhibitor in preclinical study that shows activity in diffuse large B-cell lymphoma cell lines [143].

ROS: N-Acetyl-cysteine is being investigated as an inhibitor of iNOS expression and NO production in the tumor microenvironment. In fact, a study showed positive results when evaluated the culture of T cells with N-acetyl-cysteine that seems to improve the efficacy of adoptive T-cell therapeutics [105]. Furthermore, its use in antitumor treatment is being evaluated in various clinical trials, alone and in combination therapy, due to the inhibition it exerts on prostaglandin E2, limiting its immunosuppressive effect in cytotoxic T cells [144].

Adenosine pathway: Different approaches are being studied relating to the therapeutic blockade of adenosine-mediated immunosuppression: CD39-targeting molecules, anti-CD73 monoclonal antibodies or CD73 inhibitors, and agents that target adenosine receptors. Examples of the monoclonal antibodies that target CD39 are IPH5201 [145], which is being clinically investigated alone and in combination with durvalumab or oleclumab in solid tumors in one clinical trial initiated in 2020, and SRF617, with also one clinical trial initiated at the same time. TTX-030 [146] is also an anti-CD39 monoclonal antibody being evaluated as a single agent and in combination with chemoimmunotherapy in two phase I clinical trials. Concerning the targeting of CD73, there are several monoclonal antibodies and inhibitors currently in evaluation in solid tumors, alone and in combination with immunotherapy. AB680 [147], AK119, LY3475070 are small molecules inhibitors of CD73, each of them with one clinical trial initiated in 2021, 2020 and 2019, respectively. EVOEXS21546 [148] is a dual A2AR/CD73 inhibitor under clinical evaluation. In the same way, BMS-986179 [149], CPI-006 [150], NZV930 and uliledlimab are monoclonal antibodies that target CD73 studied in phase I and II clinical trials. One of the most anti-CD73 monoclonal antibodies being studied is oleclumab [151] with around 20 phase I-II trials. Targeting the adenosine receptors is also an interesting approach to overcome the immunosuppression caused by adenosine, and that is why different A2AR antagonists, such as ciforadenant (in phase I–III studies) [152], etrumadenant (with 7 phase I–II studies) [153], NIR178 (three studies phase I–II), AZD4635 [154] (four phase I-II studies), and inupadenant (phase I study) are being evaluated, some of them with positive results [155]. CS3005 and DZD2269 [156] are A2AR inhibitors whose clinical trials were initiated in 2020. Regarding preclinical development, there are also various agents being tested: ES002 is an anti-CD39 monoclonal antibody; HBM1007 [157], PT199, and AK119 [158] are anti-CD73 monoclonal antibodies; AK123 is an anti-PD-1/CD73 bi-specific monoclonal antibody [158]; CB-708 [159], OP-5244 [160], OR-558 [161], and ORIC-533 as CD73 inhibitors [162], and AB745 [163], ARX001822 [164] and RVU330 [165] as A2AR antagonists.

Glucose: The targeting of glucose metabolism is being studied as an anticancer strategy, due to its direct effects on regulation of T cells. Metformin is being investigated because its antineoplastic effects through AMPK-mediated or AMPK-independent inhibition of mTOR; it reduces cyclin D1 inhibiting cancer cell proliferation; it also inhibits tumor cell migration and invasion by inhibiting matrix metalloproteinase-9 (MMP-9) expression. It is also notable its direct effects on preventing T cell exhaustion. For these reasons, the effect of metformin in different tumors is currently under evaluation in more than 100 clinical trials [72].

Glutamine: The blockade of glutamine metabolism by JHU083 offers a new way to overcome tumor immunosuppression by inhibiting glycolysis and oxidative phosphorylation in the tumor microenvironment. This also causes the stimulation of T effector cells and a restoration of antitumor immunity because of an up-regulation of the oxidative metabolism in T cells. This new agent was evaluated in mice models of colorectal cancer, lymphoma, and melanoma with positive results though no clinical trial is currently approved [94].

Hypoxia targeting: Targeting hypoxia is being investigated by using hypoxia-activated prodrugs or drugs that modulate HIFs in a direct or indirectly way [37]. Hypoxia-activated drugs are those inactive compounds that in a hypoxic environment are converted by one-electron reductases into pharmacologically active drugs. Some examples of these hypoxia-activated prodrugs are tirapazamine, which induces single-and double-strand breaks in DNA of hypoxic cells and is being evaluated in several clinical trials; evofosfamide (TH-302), which entered up to 27 phase I-II clinical trials as single agent and in combination with chemotherapy and showed promising activity in combination with immunotherapy in preclinical models [38], tarloxotinib (TH-4000), which is activated in the hypoxic cells within tumors into an irreversible pan-HER inhibitor and showed promising results in NSCLC [166,167] and apaziquone, an analog of mitomycin C.

Drugs that modulate HIFs that are currently being evaluated in clinical trials are OKN-007, EZN-2698; modulators of HIF-1α translation (irinotecan and topotecan which are also topoisomerase 1 inhibitors); HIF-1α degradation inducers (HDAC inhibitors such as vorinostat, panobinostat, belinostat, and romidepsin) and transcriptional activity inhibitors (PT2977 and PT2385 that which is evaluated in renal cancer) [37].

In the preclinical setting, some drugs are being evaluated: IDF-11774 [168] and RX-0047 [169] that are HIF-1α inhibitors and CP-506 [170], a hypoxia-activated prodrug.

## 4. Conclusions

Metabolism is a modulator of antitumor immune responses, either through reduced immunogenicity or increased tumor cell resistance. There are currently dozens of therapies in development to control the metabolic microenvironment that occurs in tumors and favor immune escape, with several molecules being tested in different phases of clinical trials. The knowledge of new mechanisms of immune evasion related to metabolism will allow the development of new immunotherapies, which will finally improve patients’ outcomes.

## Figures and Tables

**Table 1 biomedicines-09-00361-t001:** Therapeutic approaches targeting tumor microenvironment metabolism and examples of clinical trials evaluating the safety and efficacy of these molecules. GBM, glioblastoma multiforme; AML; acute myeloid leukemia; NSCLC, non-small cell lung cancer; HCC, hepatocellular carcinoma; GI, gastrointestinal.

Therapeutic Approaches	Drug	Mechanism of Action	Indication	Phase	NCT
IDO INHIBITION	Indoximod	IDO inhibitor	Solid tumors, glioma, GBM, AML	I–II	NCT01191216, NCT02835729, NCT02502708, NCT02460367, NCT01792050, NCT01560923,
	Epacadostat	IDO1 inhibitor	Solid tumors	I–II	NCT03516708, NCT03471286, NCT03532295, NCT02178722
	Linrodostat mesylate	IDO1 inhibitor	Solid tumors: NSCLC, GBM, melanoma, HCC	I–III	NCT02658890, NCT04106414, NCT03362411, NCT03192943,
	Navoximod	IDO inhibitor	Solid tumors	I	NCT02048709, NCT02471846
	DN1406131	IDO1/TDO2	Healthy subjects	I	NCT03641794
	EOS200271 (PF-06840003)	IDO1 inhibitor	Glioma	I	NCT02764151
	KHK2455	IDO1 inhibitor	Glioblastoma, bladder cancer, solid tumors	I	NCT04321694; NCT03915405; NCT02867007
	LY01013	IDO1 inhibitor	Solid tumors	I	NCT03844438
	MK-7162	IDO1 inhibitor	Solid tumors	I	NCT03364049
	SHR9146	IDO1 inhibitor	Solid tumors	I	NCT03491631
ARGININE METABOLISM	Tadalafil	PDE-5 inhibitor	Liver, head and neck cancer	II	NCT03785210, NCT03238365, NCT03993353
	INCB001158	Arginase inhibitor	Solid tumors and multiple myeloma	II	NCT03837509, NCT03361228, NCT03314935, NCT02903914
MCT	AZD3965	MCT1 inhibitor	Solid tumors	I	NCT01791595
GLUCOSE AVAILABILITY	Metformin	AMPK activator	Solid tumors	I–III	NCT04758000, NCT04741945, NCT04559308, NCT01243385
HYPOXIA TARGETING	OKN-007	HIF-1 alpha inhibitor	Glioma, glioblastoma multiforme	I–II	NCT04388475, NCT01672463
	Belzutifan	HIF-2 alpha inhibitor	Renal cancer	I–III	NCT03634540, NCT04736706, NCT02974738, NCT04586231
	Evofosfamide	Hypoxia-activated prodrug	Solid tumors	I–III	NCT02402062, NCT02342379, NCT02433639
	PT2385	HIF-2 alfa inhibitor	Renal cancer	I–II	NCT02293980, NCT03108066
	EZN-2968	HIF-1 inhibitor	Solid tumors	I	NCT01120288, NCT02564614
	TH-4000	Hypoxia-activated prodrug/pan-HER inhibitor	NSCLC	II	NCT03805841
	Apaziquone	Hypoxia-activated prodrug/DNA alkylator	Bladder cancer	III	NCT02563561
ROS modulation	N-Acetyl-cysteine	COX inhibitor	Solid tumors	I–III	NCT02301286, NCT02927249, NCT02804815, NCT02467582
ADENOSINE PATHWAY	IPH5201	Anti-CD39 monoclonal antibody	Solid tumors	I	NCT04261075
	SRF617	Anti-CD39 monoclonal antibody	Solid tumors	I	NCT04336098
	TTX-030	Anti-CD39 monoclonal antibody	Solid tumors	I	NCT03884556, NCT04306900
	AB680	CD73 inhibitor	GI cancer, healthy	I	NCT04104672, NCT03677973
	AK119	CD73 inhibitor	Solid tumors	I	NCT04572152
	LY3475070	CD73 inhibitor	Solid tumors	I	NCT04148937
	EVOEXS21546	A2AR/CD73 inhibitor	Solid tumors	I	NCT04727138
	BMS-986179	Anti-CD73 monoclonal antibody	Solid tumors	II	NCT02754141
	CPI-006	Anti-CD73 monoclonal antibody	Solid tumors	I	NCT03454451
	NZV930	Anti-CD73 monoclonal antibody	Solid tumors	I	NCT03549000
	Oleclumab	Anti-CD73 monoclonal antibody	Solid tumors	I–II	NCT02503774
	Uliledlimab	Anti-CD73 monoclonal antibody	Solid tumors	I–II	NCT04322006; NCT03835949
	GS-1423	Anti-CD73/TGFβ bispecific antibody	Solid tumors	I	NCT03954704
	Ciforadenant	A2AR antagonist	Multiple myeloma, solid tumors	I–III	NCT02655822; NCT04280328
	Etrumadenant	A2AR antagonist	Solid tumors	I–II	NCT03846310, NCT03720678, NCT03629756, NCT04660812,
	NIR178	A2AR antagonist	Solid tumors	I–II	NCT03207867, NCT02403193, NCT01691924
	AZD4635	A2AR antagonist	Solid tumors, prostate cancer	I–II	NCT04495179, NCT04089553, NCT03980821, NCT02740985
	Inupadenant	A2AR antagonist	Solid tumors	I	NCT03873883
	CS3005	A2AR inhibitor	Solid tumors	I	NCT04233060
	DZD2269	A2AR inhibitor	Solid tumors	I	NCT04634344

**Table 2 biomedicines-09-00361-t002:** Therapeutic approaches targeting tumor microenvironment metabolism under preclinical evaluation.

Therapeutic Approaches	Drug	Mechanism of Action	Year
IDO INHIBITION	DN016	IDO1	2018
	IACS-9779	IDO1	2019
	TQBWX220	IDO1	2018
	RG70099	IDO1/TDO	2016
	CMG017	IDO/TDO	2019
	STB-C017	IDO/TDO	2020
	EPL-1410	IDO1/TDO2	2018
ARGININE METABOLISM	AB474	Arginase inhibitor	2019
LACTATE METABOLISM	Oxamate	LDHA inhibitor	2019
	FX11	LDHA inhibitor	2015
	NHI-I, NHI-2	LDHA inhibitor	2020
	1,3-benzodioxole derivatives	LDHA inhibitor	2020
MCT	BAY-8002	MCT1 inhibitor	2018
HYPOXIA-TARGETING	IDF-11774	HIF-1α inhibitor	2018
	RX-0047	HIF-1α inhibitor	2020
	CP-506	Hypoxia-activated prodrug	2018
ADENOSINE PATHWAY	ES002	Anti-CD39 monoclonal antibody	2019
	AK123	Anti-PD-1/CD73 bispecific monoclonal antibody	2020
	HBM1007	Anti-CD73 monoclonal antibody	2020
	PT199	Anti-CD73 monoclonal antibody	2018
	CB-708	CD73 inhibitor	2019
	OP-5244	CD73 inhibitor	2020
	OR-558	CD73 inhibitor	2020
	ORIC-533	CD73 inhibitor	2020
	AB745	A2AR Antagonist	2018
	ARX001822	A2AR antagonist	2019
	RVU330	A2A/A2B antagonist	2020
GLUTAMINE METABOLISM	JHU083	Glutamine antagonist	2019

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
