# Peer review of "The Role of Metabolism in Tumor Immune Evasion: Novel Approaches to Improve Immunotherapy"

_biomedicines, 2021, doi:10.3390/biomedicines9040361_

Round 1
Reviewer 1 Report
The review article titled “The role of metabolism in tumor immune evasion: novel approaches to improve immunotherapy” by Cruz-Bermúdez et al., provides detailed and in-depth discussion of various topics such as Immunoediting, Evasion, Equilibrium, Elimination of tumor cells by the immune system. The authors further discuss the Role of Metabolism in the evasion of tumors from the immune system. In this respect the authors touch on topics such as Reduced recognition of tumors by the immune cells, where they discuss topics such as Ag-presentation by tumors, The Immunosuppressive role of tumor microenvironment, as well discussions on the Regulation of immune checkpoint PD-1/PD-L1 signaling pathway.
The authors further provide in-depth discussions on various aspects of mechanisms of Increased resistance of tumor to immune system. In this regard, the authors provide adequate discussions on various topics subjects such as: Decrease in Perforin/Grazyme activity, Decrease in IFN-γ signaling, Death-inducing receptors, Decrease in apoptotic pathways. The authors conclude their discussions by potential Therapeutic Approaches that can be employed by using various metabolic pathways.
Overall, the article is well-written, discussions are appropriate and adequate and adequate number of relevant reference citation.
Author Response
Response to reviewer 1,
Thank you very much for taking the time to review this manuscript.
English has been proofreaded in the revised version
Best regards
Reviewer 2 Report
The manuscript of Cruz-Bermúdez et al. reviews the current literature regarding the role of metabolism in tumour immune escape. The manuscript covers some key literature and will be of some use to those in the field. In addition to my suggestions below, my major concern about the paper is the discussion of the Immunoediting. With respect, this area has been covered numerous times within the literature and this is by no means the most expert attempt to summarise the process. There are numerous simplifications and the review leaves many important areas not discussed. The authors may consider shortening this component of the review and/or integrating it with their discussion of metabolism which I think will be a more valuable resource for the field.
Line 49 – Presumably this should read “immunoediting” not “immunoedition”?
Lines 80-83 suggest that immune editing and tumour-influence only occurs when a tumour is of certain size. This is not the case – tumor immunoediting can and does occur at very early stages. Please amend. (ie line 84 goes onto suggest that expression of NKG2D leads to NK-mediated killing. Why can this not happen in the blood, to a single cell? This of course IS the case for blood cancers).
Line 91 is not referenced. There is a whole body of literature to support this statement. Moreover there are numerous other chemokines released from the tumour that promote immune cell infiltration.
Lines 103-4. Please swap order of Fas/TRAIL and Perforin/granzyme. The anti-tumour impact of Fas/Trail versus Perforin/granzyme is trivial and should be reflected as such. This sentence also requires a reference.
Line 163-6 – This seems a highly contentious statement/finding. Firstly, the notion that reduced MHCI leads to reduced T cell killing seems highly unlikely as this is against the dogma. Secondly, MHCI should (and does) dramatically increase killing by NK cells as has been shown time and time again ie . PMID: 31509742. Please address this apparent contradiction of remove reference to this study.
The adenosine pathway is insufficiently discussed in the text, given that it is one of the most suppressive metabolites in the tumour microenvironment and potently impacts immune cell anti-tumour function. This is curious given that it is extensively discussed in the Section 3 as it is a key therapeutic target. Please amend this.
Author Response
The manuscript of Cruz-Bermúdez et al. reviews the current literature regarding the role of metabolism in tumour immune escape. The manuscript covers some key literature and will be of some use to those in the field. In addition to my suggestions below, my major concern about the paper is the discussion of the Immunoediting. With respect, this area has been covered numerous times within the literature and this is by no means the most expert attempt to summarise the process. There are numerous simplifications and the review leaves many important areas not discussed. The authors may consider shortening this component of the review and/or integrating it with their discussion of metabolism which I think will be a more valuable resource for the field.
First of all, we would like to thank you for taking the time to review this manuscript.
Regarding your concern about the discussion of the immunoediting process, we agree. Its purpose is not to be the focus of the review, but to serve as an introduction to put into context the aspects that metabolism can influence and are discussed in the following sections. Therefore, it is reasonable that there are numerous simplifications and areas lacking in-depth study. However, we believe that the metabolism part of the review would lose sense without this introduction, as it is useful for people outside the specific area of cancer immunology, that may use this type of review as a first approach to the subject.
Line 49 – Presumably this should read “immunoediting” not “immunoedition”?
We agree, the word has been changed accordingly.
Lines 80-83 suggest that immune editing and tumour-influence only occurs when a tumour is of certain size. This is not the case – tumor immunoediting can and does occur at very early stages. Please amend. (ie line 84 goes onto suggest that expression of NKG2D leads to NK-mediated killing. Why can this not happen in the blood, to a single cell? This of course IS the case for blood cancers).
We completely agree, we have changed the sentence to avoid mentioning the size of the tumor, as this was misleading.
- Line 79: “Tumor cells can cause alterations in the environment.”
Line 91 is not referenced. There is a whole body of literature to support this statement.
We agree, a reference was missing. We have included a new reference (13).
- Reference 13 Burkholder B, Huang RY, Burgess R, Luo S, Jones VS, Zhang W, et al. Tumor-induced perturbations of cytokines and immune cell networks. Biochim. Biophys. Acta - Rev. Cancer. Elsevier; 2014. page 182–201.
Moreover there are numerous other chemokines released from the tumour that promote immune cell infiltration.
We agree, however the purpose of that sentence is not to enumerate all the cytokines that promote cell infiltration, but to mention that some cytokines also contribute to tumor elimination through angiogenesis regulation. However, a broader set of cytokines, including the most important ones regarding immune cell recruitment and tumor elimination, have been indicated in Line 76.
- “During the development of the elimination phase different elements are involved including: cytokines (IFNα/β, IFN-γ, IL-1α/β, IL-2, IL-6, IL-12, and TNF),…”
Lines 103-4. Please swap order of Fas/TRAIL and Perforin/granzyme. The anti-tumour impact of Fas/Trail versus Perforin/granzyme is trivial and should be reflected as such. This sentence also requires a reference.
The sentence has been changed accordingly. We consider that the reference 15, used in the same paragraph, covers this affirmation.
- “CD8+ T cells can induce tumor cell apoptosis by secretion of perforin and granzymes, or by interacting with Fas and TRAIL receptors on tumor cells In addition, effector T cells express co-stimulatory molecules such as CD28, CD137, GITR, OX-40 that enhance their proliferation and survival 15.”
Reference 15: Martínez-Lostao, L.; Anel, A.; Pardo, J. How Do Cytotoxic Lymphocytes Kill Cancer Cells? Clin. Cancer Res. 2015, 21, 5047–5056, doi:10.1158/1078-0432.CCR-15-0685.
Line 163-6 – This seems a highly contentious statement/finding. Firstly, the notion that reduced MHCI leads to reduced T cell killing seems highly unlikely as this is against the dogma. Secondly, MHCI should (and does) dramatically increase killing by NK cells as has been shown time and time again ie . PMID: 31509742. Please address this apparent contradiction of remove reference to this study.
We agree with the reviewer, this was an error of reference 21 interpretation. The sentence has been corrected to:
- “However, cell lines lacking mitochondrial DNA, with reduced expression of MHC-I, showed similar sensitivity to cytotoxic T lymphocyte (CTL) mediated killing but enhanced NK mediated killing [21].”
The adenosine pathway is insufficiently discussed in the text, given that it is one of the most suppressive metabolites in the tumour microenvironment and potently impacts immune cell anti-tumour function. This is curious given that it is extensively discussed in the Section 3 as it is a key therapeutic target. Please amend this.
The text has been extended accordingly, (Page 6)
- “Adenosine: Adenosine in the tumor microenvironment deactivates the cytotoxic effector functions of NK and CD8+ T cells through A2A receptor (A2AR) signaling. In addition, adenosine polarizes myeloid cells to develop immunosuppressive phenotypes and increases the proliferation of Tregs and MDSCs (71,72). The mechanism behind the immunosuppressive role of adenosine signaling is well studied. Briefly, A2AR trigger an increase of intracellular cAMP levels through stimulation of adenylyl cyclase. In-creased intracellular cAMP levels activates protein kinase A, leading to several im-munosuppressive effects, increasing the levels of TGF-beta, IL-10, PD-1, LAG-3, and T regs, and decreasing the levels of pro-inflammatory cytokines, such as IFN-γ, TNF-α and IL-6, among others (73–75). Adenosine in the microenvironment is generated from extracellular ATP, through the consecutive activity of the ecto-nucleotidases CD39 and CD73 proteins, which are expressed on the membrane of tumor cells, as well as on stromal and immune suppressor cells, such as, dendritic cells, Th17 lymphocytes and M2 macrophages (76). Adenosine may be further degraded to inosine by adenosine deaminase. Under normal conditions, the levels of extracellular ATP are insignificant; however, under certain situations of tissue stress (inflammatory processes, hypoxia, ischemia) common in the tumor microenvironment, there is a significant increase in these levels. This ATP release can be nonspecific or mediated by different mechanisms, including the participation of exocytotic granules, plasma membrane-derived mi-crovesicles, specific ATP-binding cassette transporters and membrane channels (connexin hemichannels, pannexin 1, CALHM1, volume-regulated anion channels, and maxi-anion channels) (75). In parallel to increased ATP release, tumor microenvi-ronment conditions may increase CD39 and CD73 expression through various in-flammatory mediators such as TGF-β, IFNs, TNF, IL-1β and prostaglandin E2. In most tumor types (lung, colon, stomach, pancreas, among others) high levels of CD39 or CD73 expression are associated with a worse prognosis (73,76)”.